# Look at the Text: Instruction-Tuned Language Models are More Robust Multiple Choice Selectors than You Think

**Xinpeng Wang[1,3], Chengzhi Hu[1], Bolei Ma[1,3], Paul Röttger[2], Barbara Plank[1,3]**
[1]LMU Munich, [2]Bocconi University
[3]Munich Center for Machine Learning (MCML)
{xinpeng.wang, bolei.ma, b.plank}@lmu.de, chengzhi.hu@campus.lmu.de,
paul.rottger@unibocconi.it

## Abstract

Multiple choice questions (MCQs) are commonly used to evaluate the capabilities of large language models (LLMs). One common way to evaluate the model response is to rank the candidate answers based on the log probability of the first token prediction. An alternative way is to examine the text output. Prior work has shown that first token probabilities lack robustness to changes in MCQ phrasing, and that first token probabilities do not match text answers for instruction-tuned models. Therefore, in this paper, we investigate the robustness of text answers. We show that the text answers are more robust to question perturbations than the first token probabilities, when the first token answers mismatch the text answers. The difference in robustness increases as the mismatch rate becomes greater. As the mismatch reaches over 50%, the text answer is more robust to option order changes than the debiased first token probabilities using state-of-the-art debiasing methods such as PriDe. Our findings provide further evidence for the benefits of text answer evaluation over first token probability evaluation.

## 1 Introduction

The open-ended nature of autoregressive language generation complicates the evaluation of Large Language Models (LLMs). One popular solution to this problem is to prompt LLMs with Multiple Choice Questions (MCQs), which limit the answer space to a few candidate options, thus enabling evaluation by comparing model choices against gold labels. For MCQs, there are two main ways of extracting model choices from their generated text answers: 1) In the **text-based approach**, the choice is automatically extracted from the text answer either by pattern matching (Wang et al., 2022) or by prompting a strong LLM (Chiang et al., 2023; Li et al., 2023) such as GPT4 (OpenAI et al., 2024). However, pattern matching can often be inaccurate because each model has different response styles, requiring manual feature engineering. This makes pattern matching infeasible when evaluating many tasks and models at the same time. Using a strong proprietary model like GPT-4 as an evaluator, on the other hand, is more flexible but lacks transparency and reproducibility, and also creates high financial costs when we running large-scale evaluations. 2) In comparison, the **probability-based approach** simplifies the evaluation by ranking the log probabilities assigned to the option IDs (e.g. A/B/C/D) from the first token prediction of the model. This approach is widely adopted in many different benchmarks and LLM evaluation studies (Hendrycks et al., 2020; bench authors, 2023; Liang et al., 2022; Santurkar et al., 2023). However, recent works point to a mismatch between the text- and probability-based approaches (Lyu et al., 2024; Wang et al., 2024), showing that first token probabilities do not match text answers given by models, especially for models fine-tuned on conversational or safety data.

Moreover, recent studies have shown that the probability-based approach lacks robustness – that it is sensitive to linguistic properties of the MCQ prompt (Leidinger et al., 2023) or perturbations such as typos, adding options, word swapping (Tjuatja et al., 2023) and option position (also called selection bias, Dominguez-Olmedo et al., 2023; Zheng et al., 2023).

In this paper, we investigate the robustness of the text-based approach. We compare it with the first token answer as well as the debiased first token answer using the first token debiasing method PriDe (Zheng et al., 2023), by measuring the answer robustness under various prompt perturbations. To avoid the labor of feature engineering and the high cost of using a proprietary model as an evaluator, we fine-tune a *Mistral-7b-Instruct* model and show that it can reliably and accurately detect the choice from the text answers across models and datasets. [1]

By comparing the choice made in the text answer and the first token probability, we show that: **(1)** The text answer shows small selection bias (§3.3) and high robustness to various sentence perturbations (§3.5) across all the models we examined; **(2)** The robustness discrepancy between the first token probability and the text answer increases as the mismatch rate between them increases (§3.3, §3.5); **(3)** When the mismatch rate is high (over 50%), the text answer shows a smaller selection bias than the state-of-art first token debiasing method PriDe (§3.3). As a whole, our work provides further evidence for the benefits of text-based over probability-based MCQ evaluations of LLMs.

## 2 Mismatch between the probability and the text-based MCQ evaluation

Before the rise of instruction-tuned large language models, it was appropriate to check the probabilities assigned to the options in MCQs by the models for several reasons: 1) Discriminative models such as BERT (Devlin et al., 2019) and RoBERTa (Liu et al., 2019) are forced to only give probabilities to the available opinions by finetuning on the task with a task-specific classification head; 2) Sequence-to-sequence models such as T5 (Raffel et al., 2020) and BART (Lewis et al., 2020) are good at giving the option ID in the next-word prediction since they were specifically trained to perform NLP tasks of various formats but they are not good at "following a user's intent" in real-life scenarios (Ouyang et al., 2022); 3) Foundation models such as GPT3 (Brown et al., 2020), OPT (Zhang et al., 2022) have poor instruction-following ability which makes the text completion uninterpretable, thus, the prediction of a model is measured by ranking the log probability of the available options.

Techniques like Instruction-Tuning (Wei et al., 2021) and Reinforcement Learning from Human Feedback (RLHF) (Ouyang et al., 2022) improve the model's ability to follow the instructions and give helpful responses. Therefore, the model can directly answer the MCQs and give its choice in the text answer. This makes it less appropriate to use probability-based evaluation since the first token probability doesn't represent the text answer, even specifically asking it to start with the option ID in the prompt (Wang et al., 2024). Figure 1 shows an example of the mismatch between the first token probabilities and the text answer given by the *Llama2-7b-Chat* model. This mismatch leads to the different performance of a model on a benchmark depending on the evaluation method, as shown in Table 1.

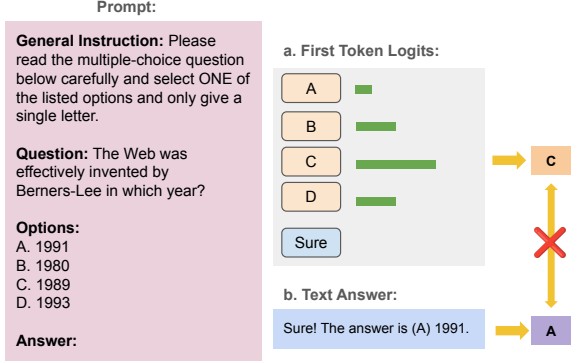

| Model (0-shot) | First Token | Text Answer |
|---|---|---|
| *Gemma-7b-Inst* | 30.2 | **50.8** |
| *Llama2-7b-Chat* | 34.9 | **43.1** |
| *Llama2-13b-Chat* | 40.2 | **47.6** |
| *Mistral-7b-Inst-0.2* | 53.2 | **53.6** |

Table 1: Performance of the models on MMLU with different evaluation methods: First token probability and Text answer. Text answer achieves better performance in general.

Figure 1: An example mismatch case between the first token probabilities and the text answer given by the *Llama2-7b-Chat* model.

---

[1]We share the dataset and trained classifiers at `https://github.com/mainlp/MCQ-Robustness`.

# 3 Robustness evaluation of the probability and text-based approaches

Several works (Dominguez-Olmedo et al., 2023; Zheng et al., 2023) have shown that large language models have selection bias such as preferring "option A", therefore they are not robust under position change of the options. To mitigate this bias, PriDe (Zheng et al., 2023) was proposed as a debiasing method to improve the robustness of the model by disentangling the token bias from the first token probability. Tjuatja et al. (2023) shows that LLMs are also sensitive to perturbation in the question such as typos, word swapping and adding more options. However, this line of work only focuses on the first token evaluation. It intuitively makes sense that the probabilistic mass on a single token is sensitive to its context and is biased towards a certain token ('A') which leads to selection bias (Zheng et al., 2023). However, it is unknown whether the text answers given by the instruction-tuned models are also sensitive to such changes. Therefore, we answer this question by comparing the robustness of the *text answers* with the answer estimated by original/debiased first token probability (referred to *first token* in the following).

## 3.1 Experimental setup

**Models**   Our study focuses on open-source instruction-tuned language models since we are interested in the text response to the user's instruction, as well as committing to open science and reproducibility. We cover three model families including Llama2 (Touvron et al., 2023), Mistral (Jiang et al., 2023) and Gemma (Team et al., 2024). We do greedy decoding during the inference time. We also trained free and robust MCQ evaluators using Mistral-7b models which we plan to make public. For simplicity, **we drop the postfix "Inst/Chat" later**.

**Benchmark**   We conduct the experiments on MMLU (Hendrycks et al., 2020) which consists of 4-option MCQs with tasks covering 57 subjects ranging from elementary mathematics to US history. Note that the original implementation of the debiasing method PriDe was done on the aggregated dataset level. In our robustness evaluation experiments, we believe it is instructive to go beyond the aggregate level: we employ and compare with PriDe on each task individually in order to gain deeper insight into the behavior of the model. We additionally use OpinionQA (Santurkar et al., 2023) inspired by Wang et al. (2024) to test the robustness of our trained classifier which will be introduced later.

**Prompting**   To give a fair comparison between the text answer and the first token evaluation, we specifically guide the model to respond directly with the option ID. We adopt the system prompt used in Wang et al. (2024) where the model is specifically asked to answer with a "single letter": *Please read the multiple-choice question below carefully and select ONE of the listed options and only give a single letter.* Our preliminary experiments show that it is important to add the constraint to respond with a single letter instead of just asking it to "choose the option" used in prior works. Without such a constraint, the model will prefer to repeat the option content instead of the option ID.

**Probability-based evaluation**   We extract the log probabilities assigned to the token corresponding to the option IDs ("A", "B", "C", "D") from the model and take the one with the highest probability as the answer. In our experiment, we see issues of only looking at the first token since different models have diverse response patterns. For example, the Llama2 models will always start the response with a space token " ", and the Gemma model starts its answer by repeating the word "Answer:" (e.g. "Answer: C") most of the time. Therefore, we also use the second token of the Llama2 model and the third token of the Gemma model, which indeed leads to slightly higher robustness results, but still lower than the text answer.

**Text-based evaluation**   To extract the choice from the text answer automatically, we follow Wang et al. (2024) to construct a classifier by training a language model. We test five different language models: *T5-small*, *T5-base*, *T5-large*, *Mistral-7b-Base-v0.1* and *Mistral-7b-Instruct-v0.2*. We finetune the Mistral models using QLoRA (Dettmers et al., 2024) implemented by Huggingface (Mangrulkar et al., 2022). See Appendix A.1 for training hyperparameters.

| Training Data | Model (Test Data) | Original Options | | | | | | Extra Options | | | | | | Acc | F1 |
|---|---|---|---|---|---|---|---|---|---|---|---|---|---|---|---|
| | | Acc | F1 | Acc | F1 | Acc | F1 | Acc | F1 | Acc | F1 | Acc | F1 | | |
| | | Mistral | | Llama2 | | Gemma* | | Mistral | | Llama2 | | Gemma* | | AVG | |
| **MMLU** (ID) | Mistral-7b-Base | 0.993 | 0.870 | 1.000 | 1.000 | 0.997 | 0.995 | 1.000 | 1.000 | 0.993 | 0.711 | 1.000 | 1.000 | **0.997** | 0.929 |
| | Mistral-7b-Inst | 0.993 | 0.870 | 0.997 | 0.874 | 0.993 | 0.992 | 1.000 | 1.000 | 0.997 | 0.997 | 0.997 | 0.997 | 0.996 | **0.955** |
| | T5-small | 0.986 | 0.987 | 0.919 | 0.913 | 0.963 | 0.965 | 0.986 | 0.570 | 0.878 | 0.665 | 0.916 | 0.380 | 0.941 | 0.747 |
| | T5-base | 0.979 | 0.980 | 0.997 | 0.997 | 0.963 | 0.964 | 0.986 | 0.667 | 0.878 | 0.665 | 0.990 | 0.664 | 0.966 | 0.823 |
| | T5-large | 0.990 | 0.870 | 0.997 | 0.997 | 0.980 | 0.860 | 0.986 | 0.570 | 0.875 | 0.482 | 0.983 | 0.663 | 0.968 | 0.740 |
| **OpinionQA** (OOD) | Mistral-7b-Base | 0.993 | 0.992 | 0.997 | 0.997 | 0.990 | 0.870 | 0.979 | 0.659 | 0.875 | 0.537 | 1.000 | 1.000 | 0.972 | 0.842 |
| | Mistral-7b-Inst | 0.997 | 0.873 | 0.986 | 0.983 | 0.977 | 0.856 | 0.986 | 0.665 | 0.878 | 0.551 | 1.000 | 1.000 | 0.971 | 0.821 |
| | T5-small | 0.973 | 0.972 | 0.959 | 0.953 | 0.786 | 0.697 | 0.986 | 0.795 | 0.871 | 0.353 | 0.920 | 0.270 | 0.916 | 0.673 |
| | T5-base | 0.986 | 0.986 | 0.986 | 0.863 | 0.926 | 0.728 | 0.979 | 0.659 | 0.875 | 0.395 | 0.963 | 0.355 | 0.953 | 0.664 |
| | T5-large | 0.979 | 0.860 | 0.993 | 0.871 | 0.980 | 0.768 | 0.983 | 0.569 | 0.864 | 0.435 | 0.993 | 0.796 | 0.965 | 0.716 |

Table 2: Performance of trained text answer classifiers. Both *Mistral-7b-Base* and *Mistral-7b-Inst* achieve good performance on test data. *We trained our classifiers only on responses from Mistral-7b and Llama2-7b models while also testing on responses from Gemma-7b.

**Annotation scheme and classifier training** The annotated data used for training the classifier is constructed as this:

**Input:** The model text response, alongside the multiple choice options.

**Label:** The ID of the correct option as determined by the model.

Table 3 is an example of the annotated model output. It is important to add references to the input since there are cases where the model gives an answer outside of the references ("No correct

Table 3: Example of manually annotated data used for training our classifier.

| Input | Sure! The least common multiple (LCM) of 4 and 10 is 40, so the answer is (C) 40. References: A. 14 B. 20 C. 40 D. 60 |
|---|---|
| **Label** | C. 40 |

answer", "Refusal", "I don't know"). We opt to annotate such cases as: {X: No correct answer, Y: Refusal, Z: I do not know}. Therefore, our classifier is required to be able to detect such cases when the model gives an answer outside of the options. In our answer floating experiments where we test if the model will change its answer with additional options, we add these three cases into the options. Therefore, we need to train two classifiers for the two cases, each of which has 7 classes.

We annotated a total of 600 samples for Gemma and 1600 samples each for the two 7b models of Llama2 and Mistral. For each model, we have the same amount of samples from two subsets: one with (**Extra Options**) and the other without (**Original Options**) additional options. See Appendix A.2 for data details. From these annotated samples, we used a total of 2000 samples for training our classifiers on Llama2 and Mistral outputs. Subsequently, we tested the classifiers on outputs from Llama2, Mistral, and Gemma models, as presented in Table 2. To test the robustness of our classifier, we also evaluate the model trained on out-of-distribution data (responses to OpinionQA) provided by Wang et al. (2024).

The result in Table 2 shows that the Mistral models provide better results than the T5 models as classifiers. The models trained on in-distribution (ID) data have near-perfect results. Even when trained on out-of-distribution (OOD) data, the performance gap in accuracy remains minimal, ranging from 1% to 4% in most cases. The successful performance of our classifier shows that it can be used in other domains as an evaluator. Furthermore, we trained our classifiers only on responses from Mistral and Llama2 models. Remarkably, when tested on responses from Gemma models, which were not included in the training data, the classifier performed comparably well, underscoring its cross-model robustness. It should be noted that the Gemma model's regular response pattern can explain the good performance on the Gemma's output. Our classifier is not optimized for extracting MCQ answers from any models, especially the ones with significantly different response styles. It is important to include diverse response styles from more model families into the training data to improve the robustness further.

## 3.2 Metrics

**Standard deviation of recalls** We follow the metric proposed by Zheng et al. (2023) to measure the selection bias by calculating the **standard deviation of recalls (RStD)** of each option. The imbalance of the recall of the option IDs indicates that the model has a selection bias towards a certain option ID. A lower RStD score indicates greater model robustness to changes in MCQ option positions.

$$\sigma = \sqrt{\frac{(r_A - \bar{r})^2 + (r_B - \bar{r})^2 + (r_C - \bar{r})^2 + (r_D - \bar{r})^2}{4}} \tag{1}$$

where $\bar{r}$ is the mean recall and $r_A, r_B, r_C$ and $r_D$ are the recall for options A, B, C and D.

**Entropy** The RStD score is based on the assumption that the correct answer is placed randomly. We introduce a second metric to assess the model's robustness by measuring the entropy of answers from multiple runs with random perturbations. For example, to test the model's robustness to option position change, we shuffle the positions of option contents N times while keeping the positions of option IDs and record the answer $A_i$ each time. The entropy is calculated as follows:

$$H(A) = -\sum_{i=1}^{m} P(A_i) \log_2 P(A_i) \tag{2}$$

where $m$ is the number of unique answers of the N runs, and $P(A_i)$ is the probability of answer $A_i$ occurring (which is the number of times $A_i$ occurs divided by N). A lower entropy means higher answer consistency under perturbation.

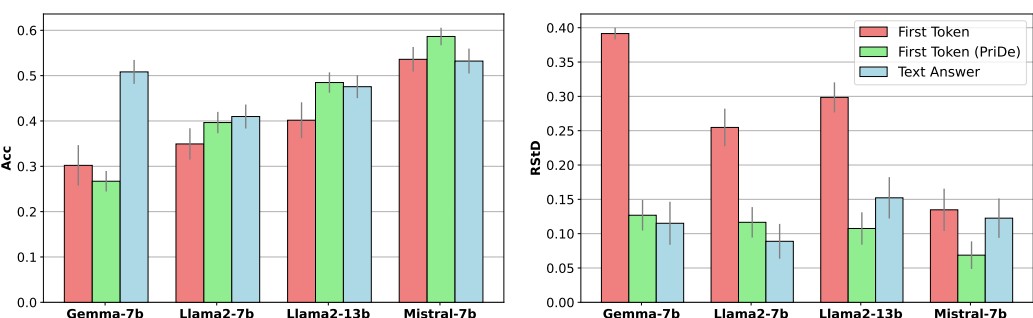

Figure 2: Accuracy and selection bias results. A lower RStD score means a smaller selection bias. As the mismatch rate decreases from Gemma (56.8%) to Mistral (10.2%), the performance gap between the first token (red) and text answer (blue) decreases. Text answers from Gemma and Llama2 have lower selection bias than the debiased first token answers.

## 3.3 Selection bias result

We compare the selection bias and accuracy of the text answer, first token answer and the debiased version of first token answer using PriDe in Figure 2. Except *Mistral-7b*, both PriDe and text answers show lower selection bias than the first token answer on all models, while text answers perform on par with or better than the debiased first token evaluation. In terms of accuracy, text answer archives higher scores than the original and debiased first token answers

| Model | Mismatch Rate |
|---|---|
| Mistral-7b-Inst-v0.2 | **10.2%** |
| Llama2-13b-Chat | 35.3% |
| Llama2-7b-Chat | 51.4% |
| Gemma-7b-Inst | 56.8% |

Table 4: Mismatch rate of first token probabilities and text outputs.

among all the models except for *Mistral-7b* where the first token answer is slightly higher than the text answer. To understand why *Mistral-7b* behaves differently, we calculate the mismatch rate between the first token and text answer as shown in Table 4. *Mistral-7b* has the lowest mismatch rate among all the models since it has a good instruction-following ability to start its first token with the option ID. Therefore, its first token answer shows

significantly lower selection bias compared to other models with a high mismatch rate and performs similarly with its text answer. Using the first token debiasing method can indeed further lower the selection bias and improve the accuracy of the model. On the contrary, *Gemma-7b* shows the highest selection bias and mismatch rate of 56.8%. Surprisingly, there is a huge performance gap between the text answer and the first token answer, even after debiasing. The text answer outperforms the debiased first token answer (PriDe) on *Gemma-7b* and *Llama2-7b* where the models have a mismatch rate over 50%. This shows that instruction-tuned language models have less selection bias than we think. In many cases, the first token debiasing method may not be necessary since the text answer is already robust, such as *Gemma-7b* and *Llama2-7b*.

## 3.4 Perturbations

To fully test the robustness of the two evaluation approaches, we further incorporate a series of different perturbation types for the prompt, based on the response bias modifications and non-bias perturbations from Tjuatja et al. (2023). Table 6 summarizes the 5 perturbation types we used. For each question, we perturbated the question four times and calculated the answer entropy except *Additional options*, where we calculated the rate of answer change. Given the evidence that the first token answer has a higher selection bias than the text answer, we need to disentangle the influence of the question perturbation and the option order. Therefore, we shuffle the option order five times for each perturbation and take the majority-voted option content for calculating the entropy, resulting in 20 runs for each perturbation type except *Option Swap*, where we study the impact of the option order itself.

## 3.5 Perturbation robustness results

**Answer consistency** Table 6 shows answer consistency results for both evaluation approaches under the perturbation *Letter Typos*, *Letter Swap*, *Word Swap*, *Option Swap* measured by entropy. Similar to the selection bias result, text answer is consistently more robust to all the perturbations on all models except *Mistral-7b*, where first token shows a minor difference compared to the text answer. The slightly higher entropy of the text answer of *Mistral-7b* can be explained by the special cases where the model claims no correct answers available or refuses to answer the question due to safety reasons (see section 3.6 for a detailed discussion), which leads to a larger option space than the first token answer. We observe that a larger model size leads to more robust first token and text answers: as the model size increases from 7b to 13b for Llama2 entropy decreases. Moreover, we note that the position of the option has the largest impact on the answer since the entropy of both approaches is the highest under *Option Swap* for all models. The robustness discrepancy between the first token and the text answer is also the largest under the *Option Swap* than the other perturbation types.

| Model | Mode | Letter Typos | Letter Swap | Word Swap | Option Swap |
|-------|------|--------------|-------------|-----------|-------------|
| Mistral 7b | First Token | 0.32 | 0.39 | 0.25 | 0.52 |
| | Text Answer | 0.34 | 0.41 | 0.27 | 0.52 |
| Gemma 7b | First Token | 0.60 | 0.62 | 0.53 | 0.87 |
| | Text Answer | **0.40** | **0.44** | **0.32** | **0.56** |
| Llama2 7b | First Token | 0.63 | 0.66 | 0.55 | 1.04 |
| | Text Answer | **0.40** | **0.44** | **0.32** | **0.68** |
| Llama2 13b | First Token | 0.53 | 0.54 | 0.48 | 0.96 |
| | Text Answer | **0.33** | **0.38** | **0.26** | **0.62** |

Table 5: Answer entropy under different perturbation types. Lower entropy indicates higher answer consistency under the perturbation. Text answer is more robust on all perturbation types across al the models except Mistral 7b where text answer performs similarly as first token asnwer.

| Original question | A multiple choice question |
|---|---|
| | ```
Question:  Which social psychological principle best explains
prejudice?
Options:
A. self-serving bias
B. in-group bias
C. individualism
D. collectivism
``` |
| **Letter Typos** | With a low probability of around 0.2, we randomly change one letter in each word of the question: |
| | ```
Question:  Which social polhagcoiyscl pinrplcie bset explains
prejudice ?
``` |
| **Letter Swap** | We randomly swap the characters of each word with a length bigger than 3 in the question, excluding the first and the last letter |
| | ```
Question:  Which social psychologicas pdinciple best explains
lrejudice ?
``` |
| **Word Swap** | We randomly swap the order of four words in the question, excluding the first and the last word |
| | ```
Question:  Which social psychological explains best principle
prejudice ?
``` |
| **Option Swap** | We randomly swap the order of choice options |
| | ```
Question:  Which social psychological principle best explains
prejudice?
Options:
A. in-group bias
B. collectivism
C. individualism
D. self-serving bias
``` |
| **Additional Options** | We add three additional options, which represent three out-of-choice options. They are: "No correct answer", "Refuse", and "I don't know". |
| | ```
Question:  Which social psychological principle best explains
prejudice?
Options:
A. self-serving bias
B: No correct answer
C: Refuse
D. in-group bias
E. individualism
F. collectivism
G: I do not know
``` |

Table 6: Pertubation types inspired by Tjuatja et al. (2023).

**Answer floating**   We also test the model's robustness to adding opinions by checking the percentage of the questions where the model changes its answer (answer floating). Figure 3 shows the answer floating result under the perturbation *Additional Options*. The first token and text answers here are majority-voted after shuffling the option orders and we use the default option order to assign the option IDs to the answers. For Llama2 and Gemma models, there is a significant percentage of questions where the model changes its first token answer after adding three more options (*Refused, I do not know* and *No correct answer is given*), reaching to nearly 70%, which is much higher than the text answer. For *Mistral-7b*, both text and first token answers achieve similar answer floating rates at around 25.5%.

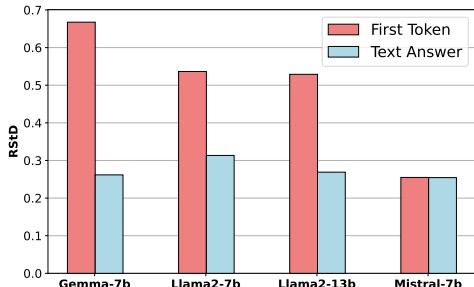

Figure 3: Answer Floating Rate. Text answers are more robust to adding options, except Mistral.

To further know about the answer floating behaviour, we zoom into the answer distribution in Figure 4. In all models, especially *Llama2-7b-Chat* and *Llama2-13b-Chat*, the first token answer shifts significantly after adding the three additional options. The number of answers shifted to the three additional options is not negligible, taking up to 14.2% of the total responses for *Llama2-13b-Chat*. After adding the options, the distribution of answers going to the three additional options is flatter compared to the text answer result. This could mean that the model selects them by chance. Again, this shows the brittleness of the first token evaluation, whose result can easily be shifted by adding new options.

Looking at the text answer result, we see a minor distribution shift after adding the options in all models. Note that the text answer can choose to "refuse" or claim "no correct answer is given" when those are not in the original options. Before adding the options, we observe different answer patterns from different model families. The Gemma model rarely gives answer outside of the options, while the Llama2 models have a high refusal rate, especially for the 7b model. The Mistral model, interestingly, tends to claim there are no correct answers more instead of refusing or showing uncertainty.

After adding the options, *Llama2-7b-Chat* tends to keep the refusal answers unchanged which is likely due to the strong safety guardrail which can be easily triggered by the keywords in the question. *Mistral-7b-Inst*, unlike other models, indeed chooses more from the added options than without them, and it is the only model that actually expresses uncertainty in the text answer such as "I do not know", which is also observed during the annotation phase.

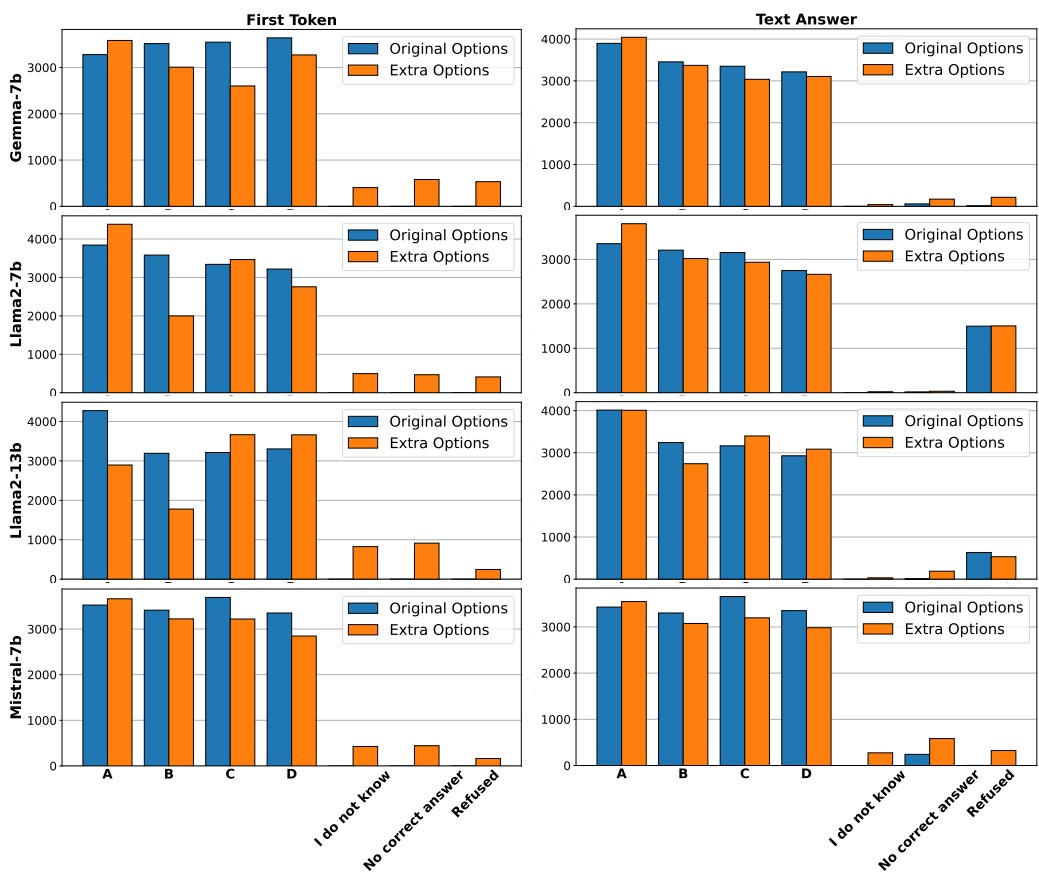

Figure 4: Answer distribution before and after adding additional options. Text answers show less distribution shift after adding additional options. Note that the text answers are not limited to the given options in the original options setting.

### 3.6 Observations

**Refusal** As we discussed in the previous section, refusal behaviour takes a large part of the total responses from *Llama2-7b-Chat*. This has a huge impact on the evaluation result as we inspect the model behaviour in each subcategory of MMLU. Figure 5 shows the selection bias of text, original, and debiased first token answers on 10 subcategories based on the top and bottom 5 subcategory performance of PriDe. A detailed overview of the accuracy scores and selection bias of responses from all models in all subcategories is shown in the Appendix A.3. Surprisingly, the text answer exhibits zero RStD on subcategory *Moral Scenarios*. As we look into the data, it turns out the model refuses to answer all the 895 questions in this subcategory due to safety reasons. This reveals the unreliability of the first token accuracy evaluation especially on sensitive topics.

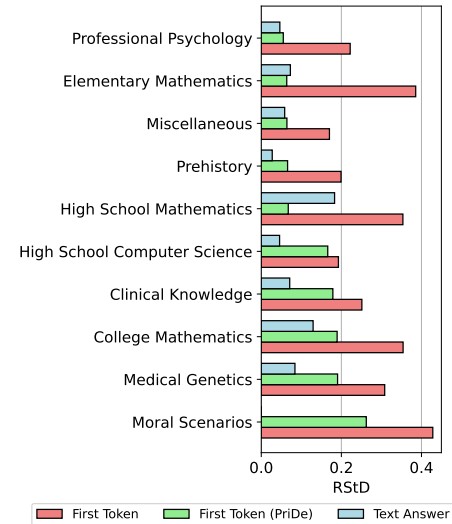

Figure 5: RStD of *Llama2-7b-Chat* in selected subcategories.

**Mismatch** We see a relation between the mismatch rate and the robustness gap between the first token and text answer. As shown in our results in Table 1, 5 and Figure 2 and 3, compared to other models, *Mistral-7b* exhibits good robustness and low selection bias, with the lowest mismatch rate of 10%. In contrast, the accuracy and robustness level gap is the largest for *Gemma-7b-Inst* which has the highest mismatch rate of 56.6%. In Figure 6, we plot the mismatch rate of the models and their first token/text answer difference in terms of accuracy and selection bias evaluated on MMLU. It shows that as the model's mismatch rate increases, the first token answer is less accurate and robust campred to the text answer. Thus, the closer the first token answers are to the text answers, the better the robustness level, whereas the robustness level of the text answer stays high. Therefore, it is recommended to always look at the text answer if the mismatch rate is unknown.

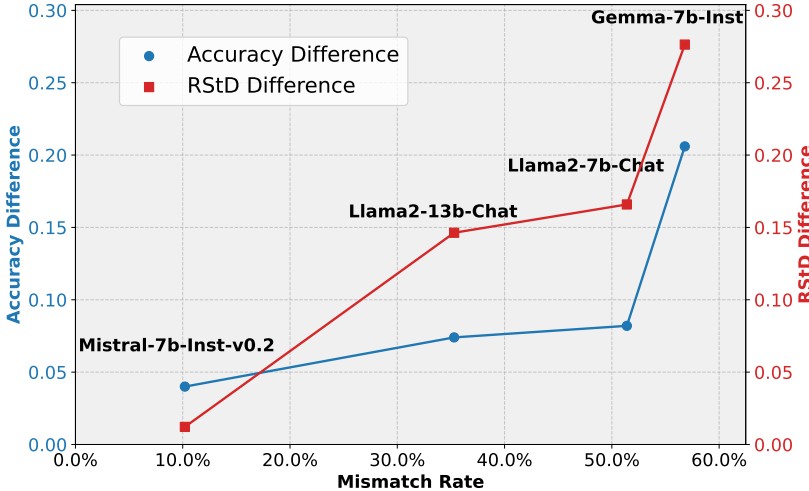

Figure 6: The absolute difference between the first token and text answer in terms of prediction accuracy and selection bias. The text answer achieves higher accuracy and lower RStd score (less selection bias) than the fist token answer across all the models. The gap increases as the model's mismatch rate increases.

## 4 Related work

**Large language models** The landscape of large language models (LLMs) has undergone dynamic evolution in recent years. With notable contributions from renowned models such as GPT4 (Achiam et al., 2023) and Gemini (Team et al., 2023), which have significantly advanced the field. Other recent advancements, including models like Mistral (Jiang et al., 2023), Gemma (Team et al., 2024), and Llama2 (Touvron et al., 2023), offer distinct settings and characteristics. These models have been equipped with innovative techniques such as Instruction-Tuning and RLHF, enabling them to effectively interpret and adhere to given instructions (Ouyang et al., 2022), and to answer with diverse and natural responses.

**Multiple choice questions evaluation** Multiple choice questions (MCQs) are widely used to assess the capabilities of LLMs across various domains. They play a crucial role in evaluation benchmarks like MMLU (Hendrycks et al., 2020), AGIEval (Zhong et al., 2023), HELM(Liang et al., 2022) and Evaluation Harness (Gao et al., 2021), as well as in evaluating moral beliefs, opinions on public issues, and surveys (Santurkar et al., 2023; Scherrer et al., 2024; Wang et al., 2024). Traditionally, MCQ accuracy has been measured based on the model's first token prediction (Dominguez-Olmedo et al., 2023; Santurkar et al., 2023). However, contemporary LLMs often offer nuanced responses, challenging the reliability of first token evaluation. This discrepancy has been revealed by recent studies (Wang et al., 2024; Lyu et al., 2024).

**Robustness** Studies by Dominguez-Olmedo et al. (2023) and Zheng et al. (2023) reveal selection bias (like 'Option A') in LLMs, but they focus solely on the first token of the model's response. Meanwhile, Tsvilodub et al. (2024) and Lyu et al. (2024) explore different evaluation methods' impact on LLM robustness, noting performance variations without a clear preference. Research on prompt brittleness, such as Röttger et al. (2024) and Leidinger et al. (2023), highlights the sensitivity of LLMs to prompt variations, impacting performance and reliability. Further investigation is required to grasp prompt brittleness's full extent and its implications for model robustness.

## 5 Conclusion

This work studies the robustness of instruction-tuned language models in the multiple-choice question answering settings. Our research builds upon previous studies which have highlighted the brittleness of first token probability evaluation and the mismatch between first token and text answer. Through extensive perturbation experiments, we demonstrate that text answers generated by instruction-tuned language models are more robust. In cases where the first token answer matches the text answer, both approaches exhibit similar levels of robustness–in our experiments this was the case only for one out of the tested models, Mistral. Our findings suggest that the instruction-tuned language models are more robust for text answers than what previously shown in works using the first token probabilities as evaluation. Therefore, we suggest to evaluate LMs in a more detailed and realistic way by directly inspecting the text answer, to have a better understanding of the full spectrum of model behaviour. We strongly caution against relying only on first token probability.

### Acknowledgments

XW, CH and BP are supported by ERC Consolidator Grant DIALECT 101043235 and in parts by Independent Research Fund Denmark (DFF) Sapere Aude grant 9063-00077B. CH is also supported by the DAAD programme Konrad Zuse Schools of Excellence in Artificial Intelligence, sponsored by the Federal Ministry of Education and Research. BM is supported by BERD@NFDI (German Research Foundation grant 460037581). PR is a member of the Data and Marketing Insights research unit of the Bocconi Institute for Data Science and Analysis, and is supported by a MUR FARE 2020 initiative under grant agreement Prot. R20YSMBZ8S (INDOMITA).

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

## A  Appendix

### A.1  Hyperparameter for the text answer classifier training

We finetuned five different language models: T5-small, T5-base, T5-large, Mistral-7b-base-v0.1 and Mistral-7b-Instruct-v0.2 for the training text answer classifier. Table 7 shows the hyperparameters and their corresponding values for the T5 models. For the two mistral models, we used QLoRA (Dettmers et al., 2024) and the default parameter settings from the Huggingface PEFT repository (Mangrulkar et al., 2022). Table 8 shows the hyperparameters and their corresponding values for the Mistral models.

| Hyperparameter | Value |
| --- | --- |
| learning_rate | 2e-5 |
| train_batch_size | 1 |
| weight_decay | 0.01 |
| bf16 | True |
| num_train_epochs | 8 |

Table 7: Hyperparameters for training T5 models

| Hyperparameter | Value |
| --- | --- |
| lora_r | 64 |
| lora_alpha | 16 |
| lora_dropout | 0.1 |
| task_type | "CAUSAL_LM" |
| use_4bit | True |
| bnb_4bit_compute_dtype | "float16" |
| bnb_4bit_quant_type | "nf4" |
| use_nested_quant | False |
| num_train_epochs | 8 |
| train_batch_size | 4 |
| gradient_accumulation_steps | 1 |
| gradient_checkpointing | True |
| max_grad_norm | 0.3 |
| learning_rate | 2e-4 |
| weight_decay | 0.001 |
| optim | "paged_adamw_32bit" |
| lr_scheduler_type | "constant" |
| warmup_ratio | 0.03 |
| group_by_length | True |

Table 8: Hyperparameters for training the Mistral models

### A.2  Annotation scheme

The annotation process was carried out by two in-house annotators, who were presented with the model outputs and the relevant multiple-choice questions. By reading the model

response, the annotator has to decide which option the model is referring to. In cases where the model contradicts itself, such as choosing 'A' followed by describing the option content of 'B', we annotate this case as a failure mode 'NaN', which is not considered in this work. After the independent annotation process, two annotators discuss the samples where disagreement occurs and resolve the conflicts. Table 9 shows the data statistics of our annotated data. It is noteworthy that in the original options setting, we never observe cases where the model expresses "I do not know", leading to 0 number of Z. However, when adding it into the options, we do often observe that model choosing "I do not know", especially in *Mistral-7b-Inst-v0.2*. The option position in the **Extra Options** setting is shuffled.

| Model | Original Options | Extra Options |
|---|---|---|
| Llama2-7b-Chat | A:168, B:163, C:183, D:171, Y:99, NaN:16 | A:126, B:152, C:151,,D:122, E:102, F:73, G:58, NaN:16 |
| Mistral-7b-Inst-v0.2 | A:223, B:131, C:248, D:141, X:20, NaN:37 | A: 156, B:96, C: 134, D:78, E:87, F: 118, G:106, NaN:25 |
| Gemma-7b-Inst | A:112, B:75, C:55, D:57, NaN:1 | A:88, B:52, C:23, D:30, E: 32, F: 38, G: 36, NaN:1 |

Table 9: Data statistics of our annotated data.

### A.3 Accuracy and selection bias in all subcategories

The detailed results the accuracy scores and selection bias of responses from all models in all subcategories are shown in the following Figures.

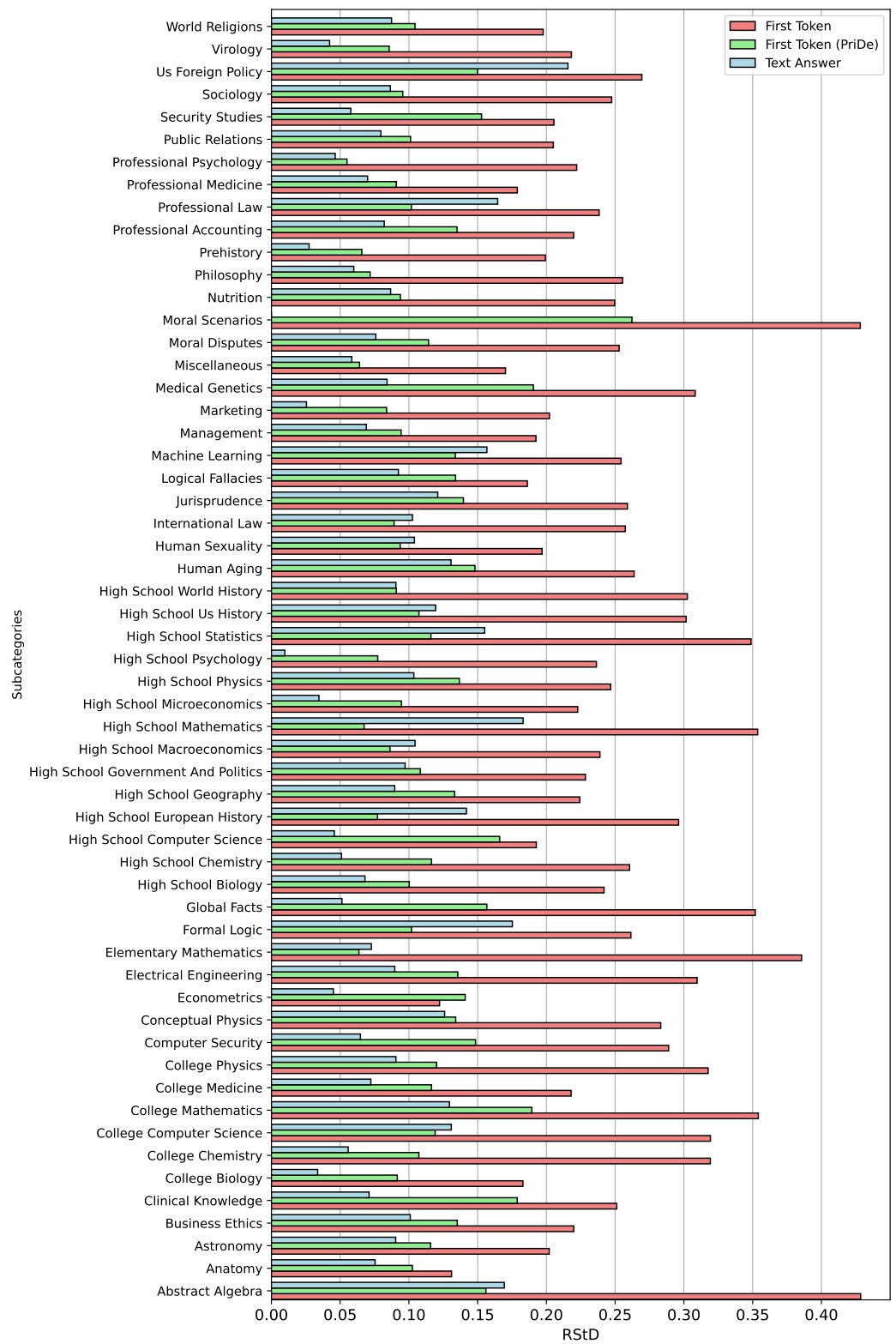

Figure 7: Selection bias on responses from *Llama2-7b-Chat* in all subcategories.

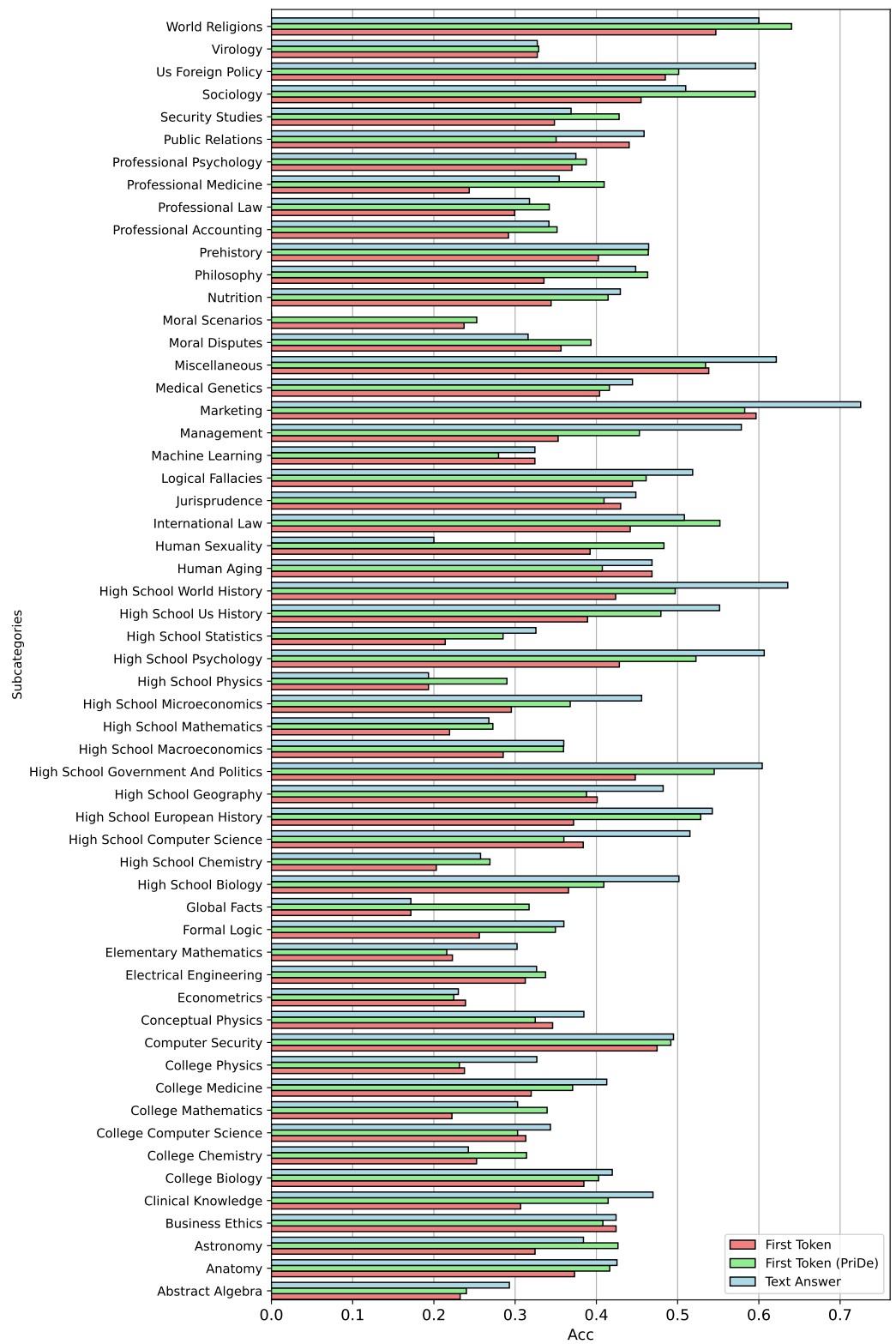

Figure 8: Accuracy on responses from *Llama2-7b-Chat* in all subcategories.

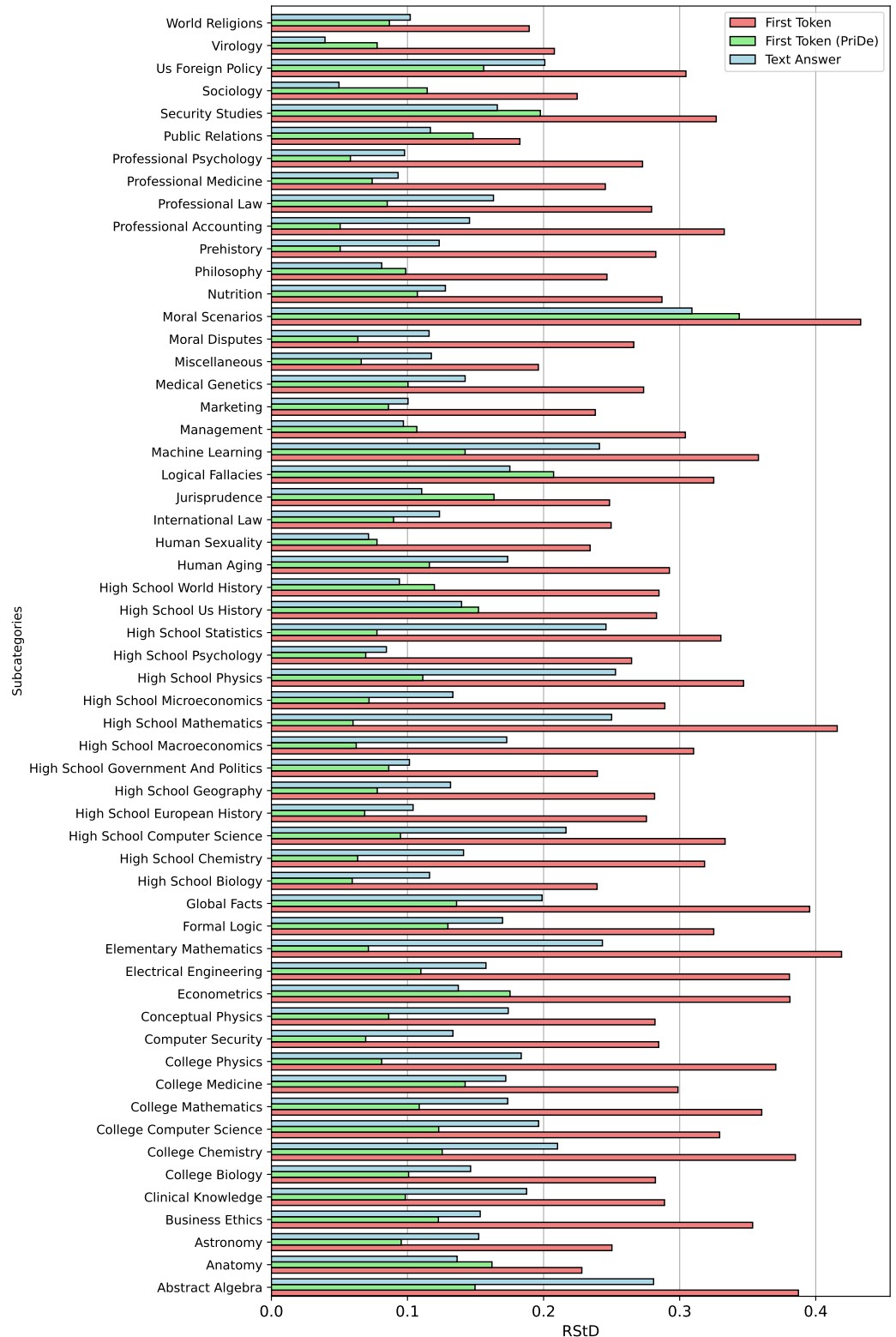

Figure 9: Selection bias on responses from *Llama2-13b-Chat* in all subcategories.

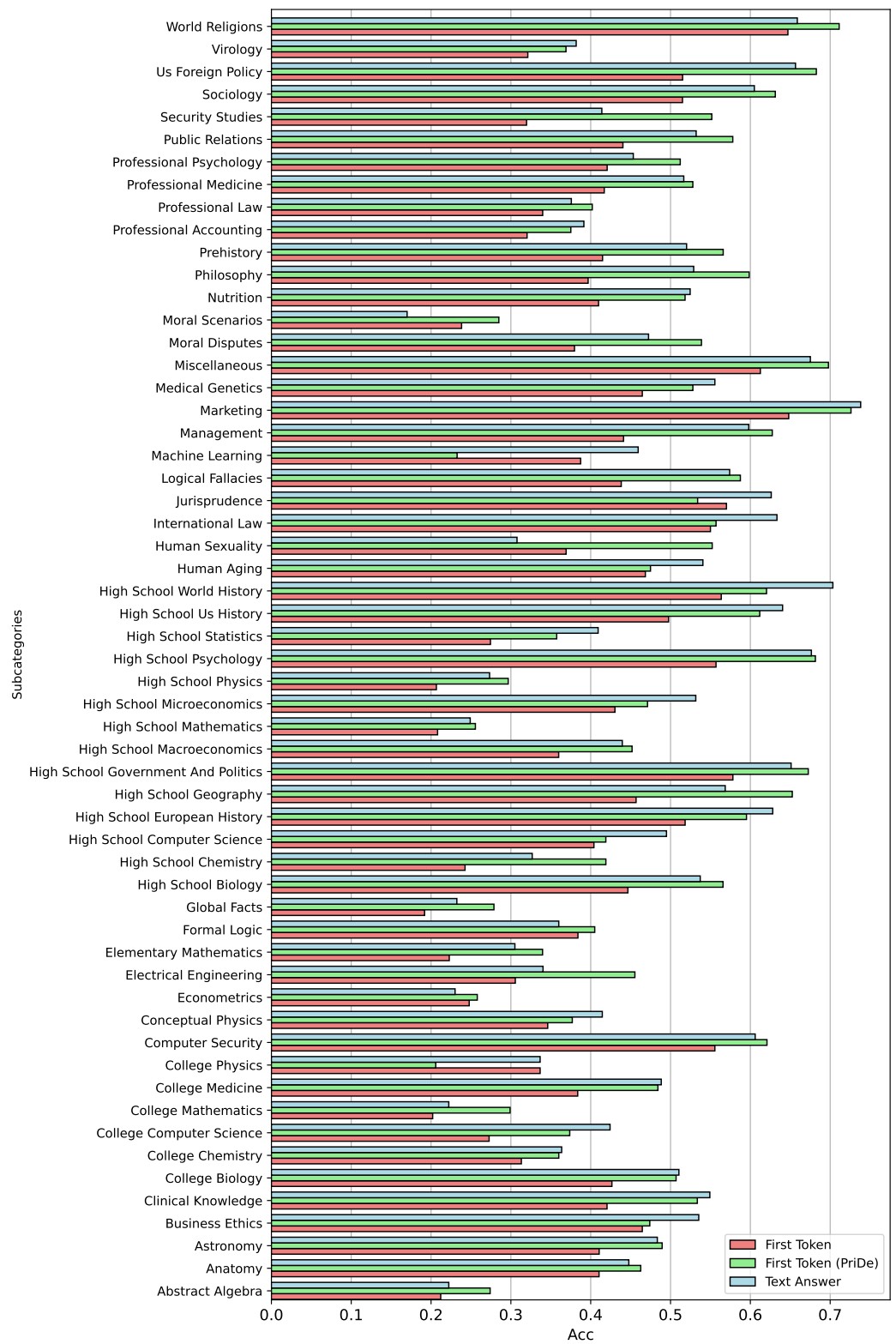

Figure 10: Accuracy on responses from *Llama2-13b-Chat* in all subcategories.

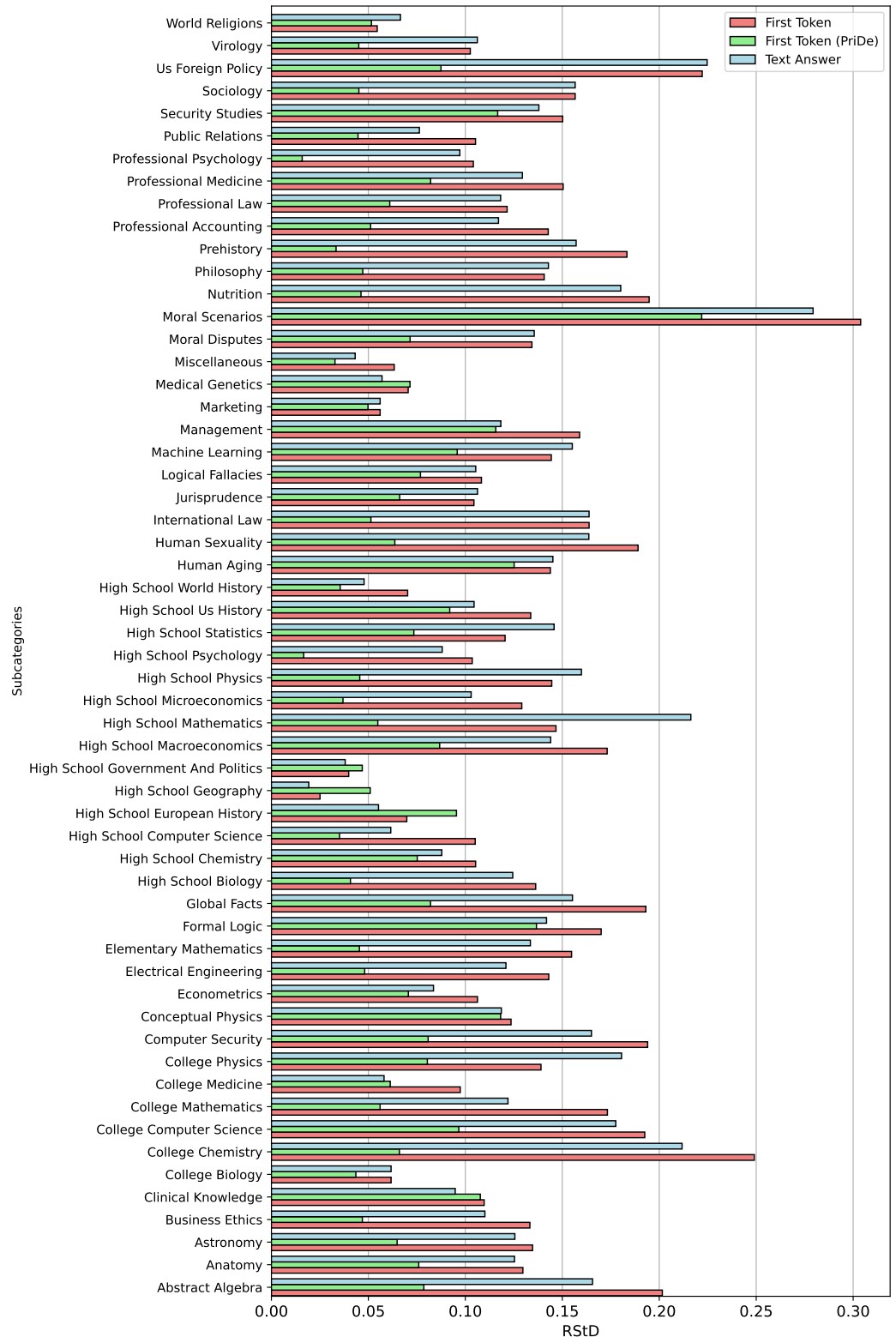

Figure 11: Selection bias on responses from *Mistral-7b-Inst-0.2* in all subcategories.

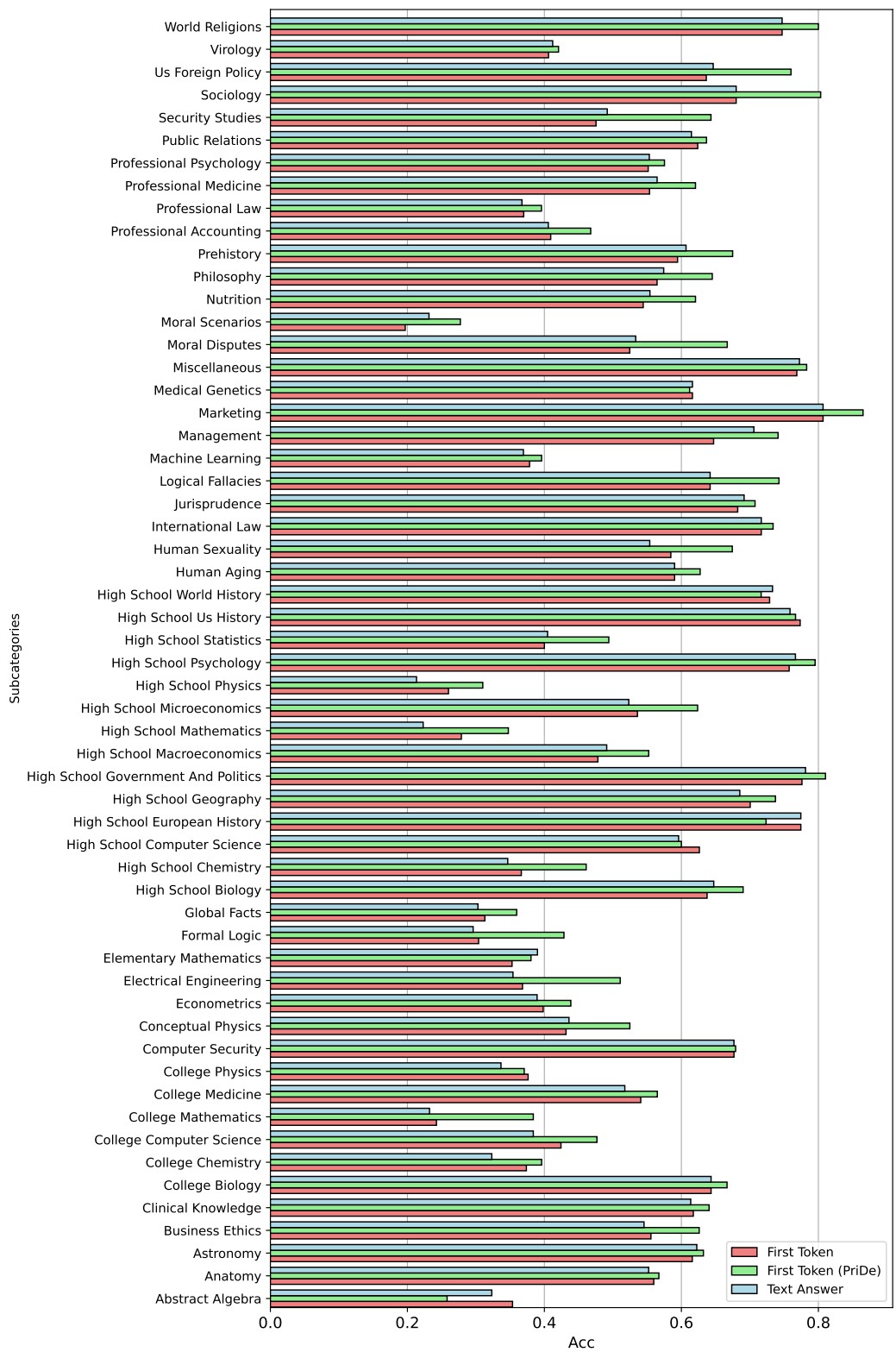

Figure 12: Accuracy on responses from *Mistral-7b-Inst-0.2* in all subcategories.

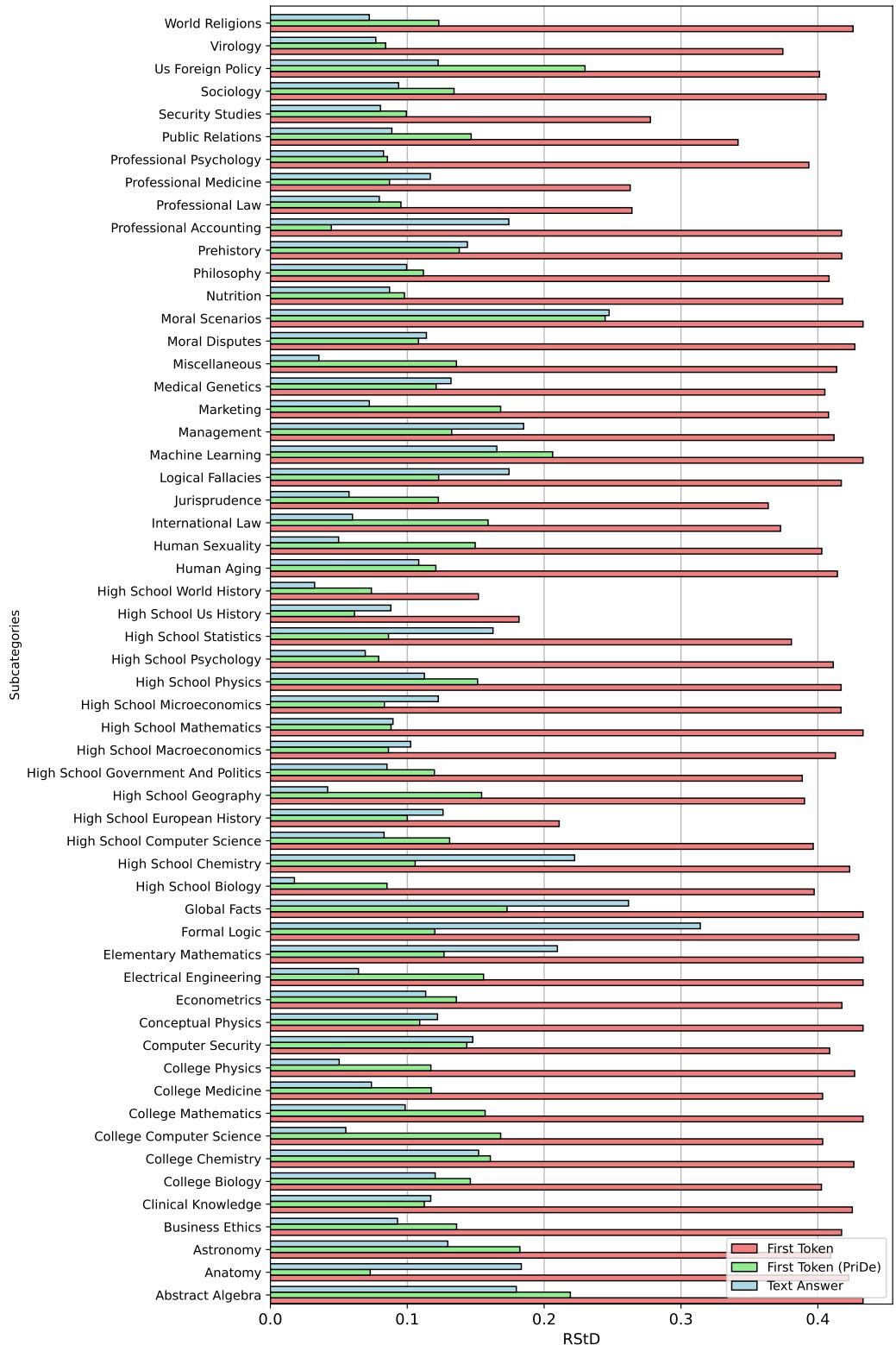

Figure 13: Selection bias on responses from *Gemma-7b-Inst* in all subcategories.

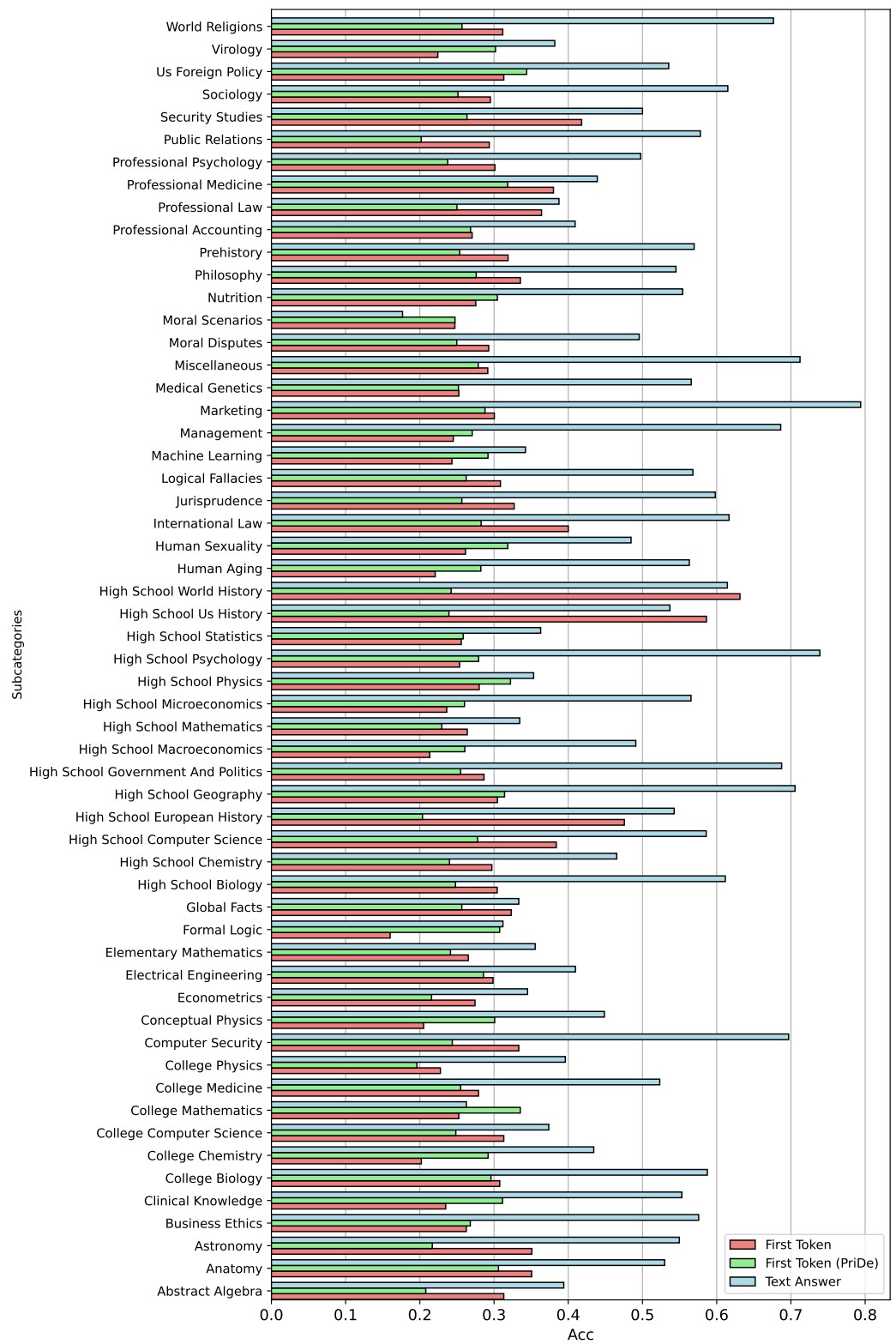

Figure 14: Accuracy on responses from *Gemma-7b-Inst* in all subcategories.

