# OpenReview forum: "Look at the Text: Instruction-Tuned Language Models are More Robust Multiple Choice Selectors than You Think"
_colmweb.org/COLM/2024/Conference — COLM_

### Official Review · Reviewer_6RFZ · 2024-05-04

**Rating:** 6
**Confidence:** 4
**Ethics Flag:** 1

**Summary:**

This paper analyses the robustness of multiple-choice QA evaluation of instruction-tuned LLMs. It compares two approaches:

1. Token probability: Ranking answer choices based on the model's predicted probability of an early token (varies by model).
2. Text-based: Extracting the answer choice from the generated text with a classifier.

The paper compares three model families (Llama2, Mistral and Gemma) on the popular MMLU task.

The authors argue that finetuning techniques have reduced the robustness when relying on token probabilities, and provide empirical evidence for this.

They provide informative experiments, and show that token probability systematically underestimates model performance on MMLU tasks. They also provide evidence for improved robustness with their classifier.

**Questions To Authors:**

* The fine-tuned Mistral model used for answer extraction could introduce its own biases, affecting the evaluation results. Have you considered measuring and mitigating this?
* Have you considered evaluating the robustness of closed, but popular models in comparison to the open models studied in the paper? (Even small samples could be interesting.)
* Do you have recommendations for finetuning methods based on your findings that could alleviate the robustness issues?

**Reasons To Accept:**

* The paper is very well written and well structured.
* The included experiments provide strong empirical evidence for the author’s case to prefer text-based methods over token probabilities.
* The findings are very relevant and important for LLM evaluation practices.

**Reasons To Reject:**

* Using a trained model for answer extraction raises concerns about the generalisability of the findings to models with significantly different instruction tuning / rlhf. Their discussion of Gemma doesn’t alleviate this, because they neither show how a model trained on Gemma data would behave, nor provide an error analysis that systematically compares the model answers. Something like this could have been achieved with a data ablation study without collecting more examples.

* The paper is very similar to ​​https://arxiv.org/pdf/2402.14499 (that they cite throughout) in both findings and methods. The main additions are 1) including the Gemma model and 2) using the MMLU benchmark instead of just OpinionQA.

---

> ### Author Rebuttal · Authors · 2024-05-30
>
> Thank you for your thoughtful feedback!
> ### **Reliability of the Classifier**
> We acknowledge the concern, however, the focus of this paper is not to design the best MCQ answer extractor applicable across all models. There are other papers focusing on creating a reliable LLM-based answer extractor for any model, which is a task beyond the scope of this paper. **Our focus is to analyse the behavior of the chosen models with the help of the classifiers we trained, which have good enough performance to be relied on, as shown in Table 3.**
>
> However, we are happy to give further analysis: the classifier needs to encounter diverse response styles to enhance its robustness. Llama2 exhibits a wider range of response patterns, which makes classifying Llama2 responses more challenging. To verify this, we compare the classifier accuracy when trained on different model responses:
> |Test data | Llama2-7b | Mistal-7b | Gemma-7b
> | -| -| -| -|
> |Trained on  Mistral-7b | 84.7 | 1.0  | 99.5|
> |Trained on Llama2-7b |99.5 | 99.5 | 1.0|
>
> Both models can generalize well to Gemma-7b because of its regular output pattern. The model trained on Llama2 can generalize to Mistral, but not the other way around.
> Therefore, it is better to include more diverse training samples.
> ### **Closed-sourced model result**
> We are happy to provide results on a close-sourced model.  Note that closed-sourced models don’t provide logits over the vocabulary, which makes token probability evaluation infeasible. We compare the text answer by asking ChatGPT-3.5 and Gemma 7b 10 questions, under the “swap option” perturbation. Both models demonstrated similar scores. **While it is a small experiment, it suggests that close-sourced models might not be more robust.**
> || Entropy |
> |-|-|
> |ChatGPT-3.5 | 0.468 |
> |Gemma-7b    | 0.459 |
>
> ### **Difference to Wang et al.[1]**
> [1] only showed the mismatch problem but did not provide a solution. We further investigated the robustness of the two approaches and showed that text-based evaluation is more beneficial. Therefore, **our paper is a direct follow-up and a solution to the problem raised by [1].**
> ### **Finetuning Suggestion**
> A straightforward solution is to add more perturbation types to the training dataset. Another solution could be using debiasing methods. Similar to PriDe, we can also design bias estimation methods to de-bias the text answer by conducting multiple runs. As PriDe uses only first token probabilities, a text-based debiasing method is required.

---

> > ### Comment · Reviewer_6RFZ · 2024-06-04
> >
> > Thank you for your response.
> >
> > **Reliability of the Classifier** Thanks a lot for adding the comparison between Mistral and Llama. The stark difference in classifier accuracy between Mistral and Llama highlights the difficulty in training reliable LLM-based answer extractor that works across models. I appreciate that solving this problem is out of scope for your paper, but I think it's important to emphasize this limitation in the paper. (e.g. in your reply, in the discussion of differences to Wang et al. you describe your approach as a solution, which seems too strong given the strong bias in this method.)
> >
> > **Difference to Wang et al.** I'm not sure what solution your paper provides that the paper from Wang et al. doesn't. If I understand correctly, that paper also trains a classifier. Can you be more specific to what the solution is your paper provides but that paper doesn't?

---

> > ### Author Response · Authors · 2024-06-04
> > **Reply to Reviewer 6RFZ**
> >
> > Thank you for your comments and further questions!
> >
> > **Reliability**
> > Thank you for your suggestions. We agree it is important to address that our classifier may not perform perfectly on models that are significantly different.  If one wants to apply our classifier to their own models, we suggest inspecting the model's response first, and further finetuning the classifier if the response style is highly different from the training data, to further improve its robustness. Therefore, we will share our classifiers so that other people can continue improving them based on their results. We will include this in the final version of the paper.
> >
> >
> > **Solution to Wang et al.**
> > Sorry for causing any confusion. By "solution", we don't mean *"how to extract the text answer"*. The solution here means *"which one to choose when facing the choice between the first token and text answer evaluation".*
> > Wang et al. showed the mismatch issue but didn't give a clear suggestion on which one is clearly better. **How much benefit do we get by doing text evaluation instead of first token probability evaluation?**
> > Given the mismatch problem shown by Wang et al, we want to know which one should we choose by comparing them systematically in terms of robustness and accuracy. In both cases, the text answer evaluation shows superior properties. Therefore, we provide an answer to the mismatch problem raised by Wang et al.
> >
> > We hope this can address your concern adequately. We are happy to give further explanations if there is still any uncertainty. If you find the revisions and clarifications satisfactory, we would appreciate your consideration in re-evaluating the manuscript.

---

> > > ### Comment · Reviewer_6RFZ · 2024-06-06
> > >
> > > Thank you for your response, the rebuttal has clarified some areas I was concerned about and I'll increase my score (and expect that a final version of the paper would also be clear about these).

---

> > > > ### Author Response · Authors · 2024-06-06
> > > > **Thank you!**
> > > >
> > > > Thank you for your valuable feedback which helps improve the quality of our paper!

---

### Official Review · Reviewer_vYwj · 2024-05-09

**Rating:** 7
**Confidence:** 4
**Ethics Flag:** 1

**Summary:**

This paper studies the robustness of instruction-tuned LLMs in multiple-choice question (MCQ) evaluation. The authors found that compared to the traditional way, where LLMs are straightforwardly prompted to predict the option IDs as answers (using the next-token prediction probability), prompting the instruction-tuned LLMs with natural instructions to select the answers leads to better robustness. More empirical analyses consolidate the claim and findings.

**Reasons To Accept:**

* This paper focuses on an important problem, i.e., the selection bias of instruction-tuned LLMs in MCQ evaluation. This is somewhat paid less attention to in previous research.
* The evaluation and analyses are thorough and sound. They involve comparison with baselines (PriDe), multiple benchmarks (MMLU and its multiple categories; OpinionQA), and multiple models. These efforts consolidate the findings in this paper and justify the effectiveness of properly prompting instruction-tuned LLMs in MCQ.

**Reasons To Reject:**

I do not see clear reasons to reject it, but I suggest the authors to polish the writing to make it more accessible. For instance, the term "mismatch" is quite unclear and confusing across abstract, intro, and Section 3.3. I think more formal definitions are needed. Also, its correlation with the robustness (Figure 2) can be better presented in scatter figures (e.g. the y-axis is a metric of robustness while the x-axis is the mismatch value).

---

> ### Author Rebuttal · Authors · 2024-05-30
>
> We truly appreciate your recognition of the importance of our work and our contribution to the community. With our findings, we encourage people to choose better methods when evaluating LLMs, especially instruction-tuned models.
>
> Thanks for pointing the writing out and we will incorporate your suggestion into our camera-ready version.
>
> **Clarifying "Mismatch"**: The mismatch rate refers to the frequency of cases where the fist token answer doesn’t match the text answer. We will add this definition to the camera-ready version to enhance clarity.
>
> **Scatter Figures for Robustness Correlation**: We appreciate the suggestion to present the correlation with robustness more effectively using scatter figures. Although we initially considered this, we were constrained by page limitations. Given that we have one additional page available for the camera-ready version, we will include the scatter figures.
>
> We thank the reviewer once again for the recognition of the importance of our work and for the valuable suggestions. We hope these modifications will enhance the clarity and accessibility of our paper and address the reviewer's concerns satisfactorily.

---

> > ### Comment · Reviewer_vYwj · 2024-06-03
> >
> > Thanks for the authors' response. I will maintain the rating.

---

### Official Review · Reviewer_k5FQ · 2024-05-10

**Rating:** 7
**Confidence:** 4
**Ethics Flag:** 1

**Summary:**

When asking a LLM to answer a multiple-choice question with, say, four answer alternatives "a", "b", "c", and "d", it is a non-trivial problem to determine whether or not the model has given the correct response. This is because the model might generate variations of the answer, like "a", "Alternative a", "Answer: a", "The correct answer is a", etc. This might happen even if the prompt to the model explicitly asks for a single-letter answer.

This paper examines various methods for determining the given answer, including first token probability (assuming that the model is well-behaved and answers with a single letter), and text-based evaluation, which amounts to training a classifier based on the text outputs from models responding to multiple-choice questions. The classes are the answer alternatives, e.g. "a", "b", "c", and "d". The investigations in the paper concern several models, datasets, and evaluation metrics. The authors have also studied the effect of perturbations, like changing the word order in the question, changing the order of the answer alternatives, adding extra answer alternatives, etc.

**Reasons To Accept:**

The paper is well-written, providing a good introduction to this research question. The experiments are systematically conducted, and the results are interesting, although not surprising

**Reasons To Reject:**

The contribution of the paper is not very big.

---

> ### Author Rebuttal · Authors · 2024-05-30
>
> Thanks for recognizing the quality of our writing and our systematic experiments. We appreciate the reviewer's positive feedback and constructive criticism.
>
>  ### **Contribution of our work**
>
> While we understand the concern regarding the perceived size of our contribution, we would like to emphasize the significance of our findings. **Many current works in the field rely heavily on the first-token probability for evaluating LLMs without being aware of the potential misalignment and robustness issue we have highlighted.**
>
> Our work addresses a critical oversight in the evaluation of LLMs, providing evidence that relying solely on first-token probabilities can lead to significant inaccuracies. This issue impacts a wide range of applications that utilize multiple-choice questions for model evaluation.
>
> Our study advocates for a more realistic and comprehensive evaluation approach by considering the entire text output with easy to fine-tune text classifier. This shift could (greatly) enhance the accuracy and relevance of future LLM assessments, leading to better model development and deployment. By inspecting the text answer, we show that instruction-tuned language models are more robust and perform better than previously shown prob-based evaluation which shows the benefit of instruction-tuning and RLHF. This raises important questions: **Does evaluating token probability on the base model really reflect its capability in real life?**  **Should we move to a more realistic evaluation framework by evaluating the text answers of the instruction-tuned models instead?** Since a large scale of users directly interact with the instruction-tuned models in the text space, it is important to move to a more human-centric evaluation paradigm.
>
> Given that one more page is allowed in the camera-ready version, we will add more discussion and implications for future works.
>
> We hope the reviewer finds our responses satisfactory. Once again, we extend our gratitude to the reviewer for their valuable suggestions.

---

### Official Review · Reviewer_4zY5 · 2024-05-12

**Rating:** 7
**Confidence:** 4
**Ethics Flag:** 1

**Summary:**

The paper tackles robustness of LLMs for MCQ answering by evaluating the actual text generated. Most work in MCQ predominantly uses first token probability, but prior work has shown that they have selection bias (Zheng et al, 2024). This paper perturbs the answer choices (editing the prompt, swapping order, and adding answers) and uses a fine-tuned model (Mistral-7b) to help evaluate text answers and measure discrepancy between first token and text answers.

**Questions To Authors:**

- Are the results in Table 6, Figure 3, and Figure 4 solely without PriDe? If so, what are the numbers on those?

- nitpicks:
  - Overall, the figures are very helpful to understanding the paper! But for some figures like Figure 2, it might be clearer to indicate a "better" direction, especially since metrics like accuracy and entropy are "better" in different directions.
  - It seems that PriDe is cited twice as a and b; is that intentional?

**Reasons To Accept:**

Experiments are clear and precise and imply that there is strong selection bias in first token probabilities, even when debiased with prior work on perturbations (3.2/3.5) and additional choices (3.5)

**Reasons To Reject:**

- Some of the experiments seem lack numbers with PriDe, which make the paper weaker; this could be a mis-reading and could be clarified in the next section.
- Some of the terms could be defined better ( especially something like "robustness discrepancy" and"mismatch rate" which are in the main claims of the paper).
- Experiments largely hinge on their fine-tuned Mistral model on human annotated texts that pick out the intended MC answer. Their results  seem strong enough to be convincing (acc generally 98%+ excepting Gemma), but some more analysis of their fine-tuned model would be helpful, especially for the few cases the model was wrong, would help quell some worries.
- The paper could be seen as less exciting with other related work like Lyu, 2024 which they cite. However, the paper goes further into depth about the discrepancy and offer a potential solution with a text-based model instead.

---

> ### Author Rebuttal · Authors · 2024-05-30
>
> Thanks for the detailed review and the recognition of our work!
> ### **Lack number with PriDe**
> PriDe targets only a single type of bias: Preferring a certain option position, which they also called “selection bias”. Therefore, it can only improve the robustness under the perturbation “Option Swap”, as shown in Fig 2.  PriDe works by gathering the first token probabilities through repeated questioning with shuffled option orders and calculating the prior probability of bias. Since we want to disentangle the influence of the option position when evaluating under other perturbations, we shuffle option orders multiple times for each question. **Therefore, PriDe will behave the same as the original token prob.** We will add more explanation in the final version.
> ### **Error Case**
> By examining the few error cases, we see that they are also hard cases for humans. We show one example:
> ```
> Question:
> Wellman cites a number of \"unpalatable implications that follow from denying a country's right to freedom of association,\" including"  A.x B.x C.x D.x
> LLama2-7b:
> I apologize, but I cannot provide a letter answer to this question as it is based on a false premise. The question asks about the implications of denying a country's right to freedom of association, but it is not possible to deny a country's right to freedom of association as it is a fundamental human right recognized by international law. Therefore, none of the options listed are valid.
> ```
> We originally labeled it as Y (Refusal) as it denies the assumption of the question in the first place. Our classifier labeled it as X (No correct answer) which also makes sense since the response says “none of the options are valid” in the end. We will provide more error cases in the final version.
> ### **Related Work**
> Both Wang 2024 and Lyu 2024 reported this mismatch issue at the same time. However, they only opened up a question but didn’t provide a solution. Our work is a direct follow-up with more focus on the robustness and a solution to the problem.
> ### **Term Definition**
> “Robustness discrepancy” is the robustness score (RstD and Entropy) difference between the token prob and text answer. The “Mismatch rate” is the frequency of cases where the first token prob doesn’t match the text answer.
> We will add them in the final version.
> ### **Figures and Citation**
> Thanks for the suggestion! We will indicate the direction for clearer understanding.
>
> The double citation was a mistake. Thanks for pointing that out!

---

> > ### Comment · Reviewer_4zY5 · 2024-06-04
> >
> > Thank you for the clarification. I will retain the current rating.

---

### Comment · Area_Chair_DDpz · 2024-06-03
**Author response**

Hi reviewers,
Please take a look at author's responses and other reviews of the paper. If the rebuttals addressed your concerns, please let the authors know about this and update your review. If not, please continue to engage with the authors and the other reviewers in the discussion forum.

Overall, most of the reviewers are positive about this paper.

Reviewer 6RFZ:
- One of your concerns is that the generalizability of the results to other closed models or those trained with other instruction/rlhf training. Does the author response address your concerns?

Reviewer 4zY5 also raised similar concerns. Other reviewers should also feel free to weigh in on the above points.

Thanks!

---

### Decision · Program_Chairs · 2024-07-10

**Decision:**

Accept

**Comment:**

This paper studies the robustness of instruction-tuned LLMs in multiple-choice question (MCQ) evaluation. The authors found that compared to the traditional way, where LLMs are straightforwardly prompted to predict the option IDs as answers (using the next-token prediction probability), prompting the instruction-tuned LLMs with natural instructions to select the answers leads to better robustness. More empirical analyses consolidate the claim and findings.

The evaluation and analyses are thorough and sound. They involve comparison with baselines (PriDe), multiple benchmarks (MMLU and its multiple categories; OpinionQA), and multiple models.

The writing of the paper could be improved, specially to distinguish better from prior work (Wang et al.)